# A Retrospective Assessment of Temperature Trends in Northern Europe Reveals a Deep Impact on the Life Cycle of *Ixodes ricinus* (Acari: Ixodidae)

**DOI:** 10.3390/pathogens9050345

**Published:** 2020-05-01

**Authors:** Agustin Estrada-Peña, Natalia Fernández-Ruiz

**Affiliations:** Department of Animal Pathology, Faculty of Veterinary Medicine, University of Zaragoza, 50013 Zaragoza, Spain; natalia.ferru@gmail.com

**Keywords:** *Ixodes ricinus*, trends of temperature, impact, life cycle, Northern Europe

## Abstract

This study modelled the changes in the development processes of the health-threatening tick *Ixodes ricinus* in Northern Europe as driven by the trends of temperature (1950–2018). We used the ECA&D dataset to calculate the annual accumulated temperature to obtain the development rates of the oviposition, incubation, larva–nymph, and nymph–adult molts. Annual values were used to ascertain the trend in development rates of each stage. The ecological classification of Northern Europe (LANMAP2) was used to summarize results. The temperature in 1950–2018 clearly increased in the target territory. The development rates of every tested life cycle process were faster along the time series. Faster oviposition and incubation rates resulted in central Sweden, Baltic countries, and parts of Finland. Faster molting rates were observed in the same territories and in large areas of Western Norway. The trend of temperature in the period 1950–2018 shows a consistent inflection point around 1990, demonstrating that the increased annual accumulated temperature has a deeper impact on the life cycle of *I. ricinus* since approximately 1990. Faster development rates could be part of the processes driving the reported spread of the tick in the target area and should be considered as a serious threat to human health.

## 1. Introduction

World scientists recently released a “Second Warning to Humanity” [1] based on the manifesto issued 25 years ago by the Union of Concerned Scientists. Ripple et al. [1] promoted a “call” about the losses of natural landscapes, the progressive deterioration of the landscape, the exhaustion of resources, and, in general, a scientific view of the effects of humankind on climate and other natural resources. It is widely accepted that in Europe climate tends to be warmer, with shorter autumns and winters, and with more unpredictable rainfall patterns, impacting in yet unknown ways the life cycles of many organisms. While studies mostly addressed the impact of the climate trends on free living organisms, there is a lack of information regarding the impact on the life cycle processes regulating the development of ticks of potential importance for human health, because of its role in the transmission of pathogens to humans.

Studies addressed the predicted impact of climate trends on species of plants or animals, commonly using mechanistic models matching known distributions with explanatory variables of diverse quality [2,3,4]. Some of these studies explicitly addressed the expected effects of the climate trends on the distribution and phenology of parasitic arthropods [5,6]. Further research focused on the modelling of life processes of mosquitoes [7,8]. The interest on ticks increased in the last years after the realization that human pathogens are (re)emerging, like the agents of Lyme borreliosis [9,10] or the recurrent epidemics and apparent spread of Crimean–Congo hemorrhagic fever [11,12].

Ixodid ticks are ectothermic organisms with a complex life cycle, in which as many as three different stages quest in the vegetation looking for a new host for feeding. After the blood meal, the tick detaches and molts within the humid places of the ground. Therefore, a change in temperature implies an impact on the development rates of the tick. A faster development rate does not immediately imply an increased abundance of ticks, since the presence and abundance of suitable hosts play a role in tick abundance [13]. Temperature drives the development rates of ticks, and relative humidity outlines their mortality rates, either while molting or questing for a host. It has been demonstrated that cold winters may “reset” the population of ticks, greatly reducing the populations of these arthropods [14,15]. It has been pointed out that shorter and warm winters impact ticks and hosts and are part of the chain of events leading to larger contact rates between humans and ticks [16,17].

The spread of the tick *Ixodes ricinus* towards Northern European latitudes is of concern, as is the case in Southern Canada with *Ixodes scapularis* [18,19,20]. The spread of *I. ricinus* and transmitted pathogens in both latitude and altitude has been already documented by field surveys [21,22,23,24]. However, it has been pointed out that climate is not the only cause driving the changing distribution patterns of the tick in Europe, since the availability of suitable hosts may be also of central importance [25,26]. Tick females fed on large ungulates could oviposit literally thousands of eggs, resulting in large populations of larvae the next year, which could contribute to an increased density of the tick. It has been thus suggested that the (re)introduction of wild ungulates in areas of Northern Europe might interact with the warmer climate producing the observed changes of the distribution and density of the tick [27,28].

This study modelled the impact of the changes in temperature on the development rates of the tick *I. ricinus* in its northern distribution fringe, the area expected to be more affected by the trends of climate. We used a long series of interpolated daily values of temperature over Europe for the years 1950–2018 to explicitly evaluate (i) the rate of change in temperature in the period, and (ii) the effects of the changing temperature on the duration of the development periods of *I. ricinus*. We also evaluated the variability of the impact of temperature on the tick’s life cycle in different ecological divisions of Europe aiming for a comparative background along the large geographical divisions of the target territory. Results are expected to provide an adequate framework for both adaptation and mitigation in human health.

## 2. Results

### 2.1. Temperature Increased in the Target Region in the Period 1950–2018

All the results were referred to the description of ecological regions of Europe according to the LANMAP2 scheme (see Figure 1 for an explicit description of the target territory). The analysis of the annual values of temperature in the time series 1950–2018 demonstrated a clear increase of the temperature in the target territory (Figure 2). The increase of temperature was smaller in areas of Ireland, Western and Northern United Kingdom, and South-Western Norway. The greatest increase was observed for a wide territory eastern to Baltic countries, North-Eastern Sweden and northern Finland. The large territory in Russia (Figure 2) with a slope of “0” resulted from the lack of interpolated climate data, but still included in the maps because it is part of the LANMAP2 ecological description of the territory. No slope less than 0 (which would indicate a trend to decrease of temperature) has been observed.

Table 1 summarizes data for seven ecological regions in the target territory (the category “Arctic” has not been included because the scarcity of the recording stations producing poorer data). The Table 1 also includes percent of change around the break point in the trend of temperature that has been consistently observed around the year 1990. We therefore calculated the mean annual accumulated temperature separately for the periods 1950–1989 and 1990–2018, with an estimation of the percent of the difference. For every ecological region, the mean accumulated annual temperature is always higher in the period 1990–2018, at a variable percent depending on the latitude: northern regions experienced a more abrupt change.

### 2.2. The Development Rates of I. ricinus Life Cycle are Faster Mainly in Northern Countries

The values of the development rates of every life cycle stages of *I. ricinus* had a negative slope in the whole target territory, meaning for a trend to being shorter throughout the time series. No regions with a positive slope (i.e., an increase of development rates) were found, minimum being in areas where the tick is recorded to be absent. However, the different life cycle stages were not affected in the same magnitude by changes of climate as shown by different slopes along the period of study (Table 2).

All the ecological regions displayed negative slopes, meaning for a decrease of the development periods. The Figure 3 summarizes the results in the spatial extent. Moderate to high increase of oviposition rates were found in central Sweden, parts of Finland, and adjacent territories of Russia. The impact of climate on the incubation rates is more noticeable: areas with a faster incubation cover large regions of Central and Southern Sweden, Southern Finland, most of Baltic countries and small adjacent territories of Russia for which data are available. The changes of the larva–nymph molt rates show a similar geographical distribution, but the effects of the climate on this stage extend far Southern Sweden and are more noticeable in Southern Finland and adjacent Russian territories. A faster nymph–adult molt has been observed in most of the explored territory. The impact is maximum in Central Sweden and Finland (largest negative slope). It is the only life cycle stage that had obvious changes in the complete period of time in coastal Norway.

The slope of the development rates gives a general overview of the trend in a long time period. Since an obvious and abrupt change of slope between the years 1950–1989 and 1990–2018 was detected, the differences in the development rates averaged for both periods of time (in %) were calculated. The results are summarized in Figure 4 (along the gradient of Latitude) and in Table 3 (summarized by ecological region), with data shown as the percent of change (i.e., a change of 100% means for a double development rate or, in other words, for half the time in stage development). The results point to rates of change of about 74–90% faster in large areas of Central Europe, United Kingdom, Ireland, coastal Norway, Southern and Central Sweden, Southern Finland and the Baltic countries.

## 3. Discussion

This study modelled the effects of the changes of temperature in Northern Europe on the development rates of the tick *I. ricinus* in the period 1950–2018. Like any other ectothermic organisms, ticks are deeply affected by changes of temperatures. Warmer and shorter winters have been pointed out as one of the factors driving the colonization of the tick at northern latitude in Europe [27]. Studies (i.e., [25,27,28]) have all confirmed the spread of the tick in areas where it was absent only 15-20 years ago, according to field surveys at the time. Climate and vegetation impact tick survival because (i) they are determinants of the occurrence of suitable communities of hosts, and a refuge protecting ticks for off-host desiccation, (ii) host-seeking activity is affected by ambient temperature and humidity, and (iii) rates of development of ticks from one life stage to the next depend on temperature, being faster at higher temperature.

We used the interpolated ECA&D dataset of temperature for calculating the accumulated annual temperature and evaluating the rates of development of the tick’s processes for each year. We promoted the use of satellite imagery as explanatory variables for mapping the predicted distribution of ticks [29] but the relatively short series of data available from different satellites prevents large periods of time to be evaluated. Since this is a retrospective study, we aimed to have a long climate series. Only interpolated climate datasets have the length necessary to capture the general patterns of change in the life processes of living organisms.

This study could not use the basic reproduction number, R_0_, which is the universally accepted value translatable to all branches of epidemiology including those involved in studies of arthropod vector dynamics [30]. This was due to the lack of water vapor data in the climate dataset; therefore the saturation deficit (affecting survival of the ticks) could not be used to address the mortality of the tick. The focus was paid only on the impact of the temperature on the development rates of *I. ricinus*. We however assumed, as is the case of *I. scapularis* in Southern Canada, that the low water deficit in the region is not responsible of large mortality rates of the tick [28,30]. Previous studies describing the changes of environmental suitability for *I. ricinus* in Europe suggested that temperature is the limiting factor for the establishment of the tick in northern latitudes, while depletion of water vapor in the air is one of the limiting factors in its southern distribution limit [27,31]. Results were conclusive: Northern Europe is experiencing the largest impact of the climate on the development rates of *I. ricinus*. These results match well the published reports [25,27,32] regarding the spread of the tick into European northern countries. It however must to be noted that temperature (and other environmental factors) could most probably affect also the distribution and activity of the vertebrate hosts, resulting in potential changes of tick-host contact rates, of yet unknown consequences.

Time series analysis detected a strong signature of change in the temperature trend in Europe around the year 1990, which is consistent across the complete geographical range examined. The number of climate recording stations used for both periods of time (pre- and post-1990) was approximately the same and the interpolation procedures were the same. Therefore, it should be assumed that a change of the trend of temperature in Europe existed around 1990, pointing to a sharper increase of temperature noticed in the complete region. This break makes two clear periods of temperature increase, and its impact is thus different according to the regions of the target area, changes being most noticeable in the northern range of the region studied.

The effects of such changes in the tick’s life cycle cannot be ignored in the context of human health. Nevertheless, the complex tick’s life cycle and the variability of tick-host contacts, makes a straightforward interpretation difficult. Faster development rates in an environment in which water deficit is a less significant driver of tick mortality would probably denote higher tick density. Since this has been noticed for a territory like Northern European countries in which large ungulates are common, blood source for adult ticks seems to be easily accessible [33,34]. A faster tick’s life cycle would also probably alter the seasonality of the tick: this could result in changes of the rates of tick-transmitted pathogens to humans in the area, at a yet unevaluated rate [35,36]. These challenges to public health systems in Northern Europe must to be adequately addressed for the expected impact.

## 4. Material and Methods

### 4.1. Purpose

We aimed to capture the impact of the temperature in the period 1950-2018 on the temperature-dependent processes of the life cycle of the tick *I. ricinus* in Northern Europe. The date is the oldest for which available daily climate data covering the European territory exist at an adequate spatial resolution (0.25°). This study is not intended to evaluate the “abundance” or “density” of ticks in a territory, or to calculate the complete life cycle of *I. ricinus* in the target area. No humidity data are available at this resolution, and it is known that water stress has an important impact on the tick’s mortality. The length of the questing period and its implicit mortality could not be calculated because it depends on the density of available hosts, a feature that has a local nature. The scarcest the host, the longest the time of questing, and then the highest the water stress and the mortality. The lack of data on host’s density and water saturation deficit would make unreliable the evaluation of the tick’s density and mortality. Therefore, this study is restricted to (i) evaluate how the temperature-dependent life cycle stages of the tick performed in a period of 69 years, and (ii) to model how these changes evidenced in the different ecological divisions of the target territory. The study is restricted to the northern portions of Europe, where the largest impact of climate should be expected.

### 4.2. Data on Climate and Ecological Regions in Europe

The climate data were obtained from ECA&D (acronym for European Climate Assessment & Data), which is a dataset of in-situ meteorological observations within Europe gridded into a geographic projection (currently at https://icdc.cen.uni-hamburg.de/1/daten/atmosphere/ecad/ accessed, September 2019). Original climate data include temperature, rainfall and pressure at sea level, at a daily time resolution and 0.25° of spatial resolution [29]. Only average daily temperature data for the period 1950–2018 were used, since rainfall is not an adequate proxy of water vapor or relative humidity [30]. Data were imported from its original netCDF format into the R programming environment [37] for further analyses. The calculation of the development rates of the tick for multiple ecological regions on a daily basis would be unrealistic and difficult to compare in a period of 69 years. We thus opted for the calculation of the accumulated degrees Celsius in one complete year, as the input for the equations calculating the life cycle.

We adhered to the scheme of ecological regions of Europe outlined in LANMAP2 [38]. The European Landscape Map, LANMAP2, is a pan-European landscape database at a scale of 1:2,000,000. LANMAP2 covers an area of approximately 11 million km^2^ and is a hierarchical classification of 350 landscape types (more than 14,000 mapping units with an average size of 774 km^2^. This study is spatially restricted to the coordinates 66°N, 25°W (top-left) and 47°N, 40°E (bottom-right). Figure 1 includes the standard definitions of the target territory.

### 4.3. Calculation of the Development of Ixodes ricinus

We evaluated the annual duration of four development periods of *I. ricinus*, namely (i) the pre-oviposition plus oviposition periods, herein referred as “oviposition”, (ii) the incubation of the eggs (from the end of oviposition to larval hatch), (iii) the duration of the larva–nymph molt, and (iv) the duration of the nymph–adult molt, individually for the years 1950–2018. We acknowledge that this does not cover the complete life cycle of the tick, but the study is aimed to interpret the impact of the temperature on target stages of the life cycle. All the equations used to calculate the development rates of the four stages listed above were previously published [39]. The length of the oviposition, incubation, larva–nymph, and nymph–adult molts was calculated for every year and for every single 0.25° cell for which data exist in the target territory, using the accumulated annual temperature of each single cell.

### 4.4. Other Calculations

Aiming at summarizing, all the data obtained for every single 0.25° cell were transferred into the polygons representing the European ecological regions outlined in LANMAP2. Each polygon was loaded with the median value of either the accumulated temperature or the development rates, for each year. We calculated the trends of either the accumulated annual temperature or the development rates with functions available in the package “greenbrown” [40] for the R programming environment [37]. The trend was calculated as the slope of values for the time series between the years 1950–2018, using the function ‘trend’ and the method “SeasonalAdjusted” explicitly addressing an annual seasonal cycle. For every ecological region, we did plot the trends of the accumulated yearly temperature and the development rates of each stage (in days) for the period 1950–2018.

We examined if the series of temperature data had statistically significant break points. A break point is a moment of the complete time series were the data suddenly change its trend. A break point is an abrupt change in the series, in which data shows an abnormal and sustained behavior different form the one observed in a previous period of time. We evaluated the break points of every time series (accumulated temperature and the four development rates) using the package “changepoint” [41] and the function “cpt.meanvar”, available for the R programming environment [37]. After calculating the existing breakpoints for the temperature series of each single 0.25° cell, results were transferred into the ecological regions of the target territory for summarizing.

## 5. Conclusions and Future Directions

Temperature has changed at an unprecedented rate in Europe in the last decades, the slope of increase after the year 1990 being clearly higher than for the period 1950–1989. These trends led to large rates of change of the development processes of the tick *I. ricinus* in regions of Northern Europe. The application of a process-based model identifying the changes of the tick’s life cycle demonstrated that oviposition, incubation, and molting processes have been impacted resulting in faster development rates. While the presence of adequate hosts for tick feeding is a further element affecting the establishment of permanent populations of *I. ricinus*, the changes of temperature affected the progression of the tick population at its northern fringe. The joint spread of the tick vector with expected similar event of pathogen’s vertebrate reservoirs will introduce emerging pathogens into areas that were free of them until a few years ago. It is necessary to prepare an adequate framework evaluating the impact on human health.

## Figures and Tables

**Figure 1 pathogens-09-00345-f001:**
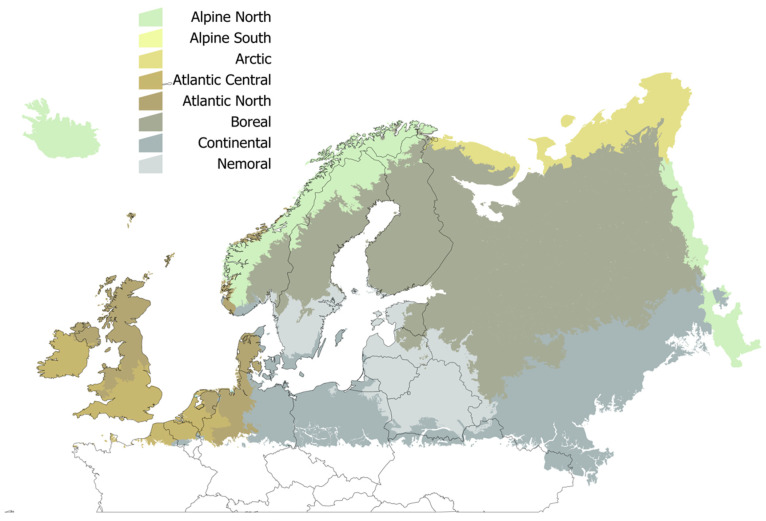
The spatial distribution of the ecological regions in the target territory, according to the standard denominations of LANMAP2. Colors are random and used only to separate the regions.

**Figure 2 pathogens-09-00345-f002:**
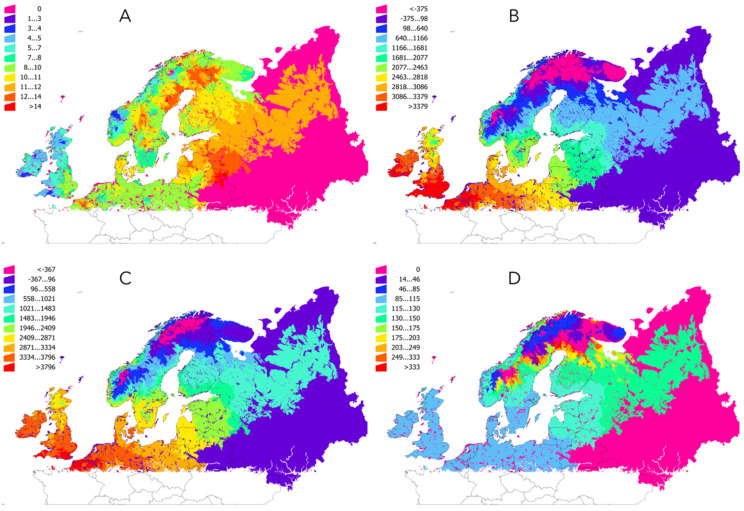
Changes in temperature in the period 1950-2018. (**A**) The slope of the changes of temperature of the complete period. The large territory in Russia with a homogeneous value of “0” are areas where no temperature data are available. Minute areas of the territory across Europe with a value of “0” are polygons smaller than the resolution of the layers of temperature (no data). (**B**) The average annual accumulated °C in the period 1950–1989. (**C**) The average annual accumulated °C in the period 1990–2018. (**D**) The ratio of change in temperature between the period 1950–1989 and 1990–2018 (a change of 100% means for twice the temperature between 1950–1989 and 1990–2018).

**Figure 3 pathogens-09-00345-f003:**
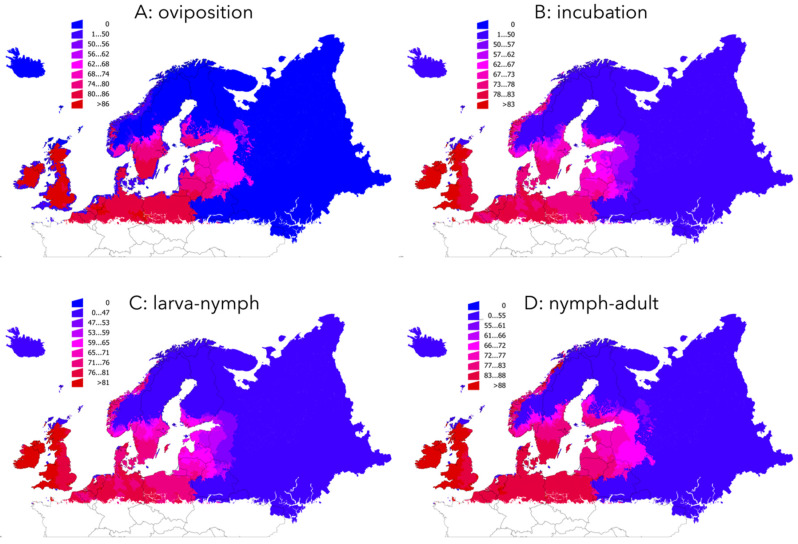
Changes of the development rates of four processes of the life cycle of *Ixodes ricinus* in the period 1950–2018. (**A**) oviposition; (**B**) incubation; (**C**) larva–nymph molt; (**D**) nymph–adult molt. Data are presented as % of change (a 100% change means twice the speed of the process).

**Figure 4 pathogens-09-00345-f004:**
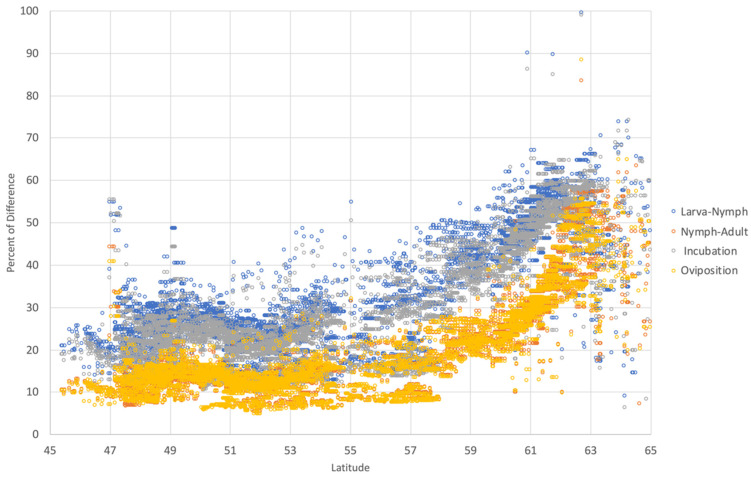
The percent of change of development stages of *I. ricinus* in the period 1950–2018 according to the latitude of the target region. Each dot corresponds to the value of each single cell of 0.25° of resolution over the complete target territory in Europe. All the rates of development are higher at northern latitudes (above approximately 58°N).

**Table 1 pathogens-09-00345-t001:** Values of the annual accumulated temperature (°C) in the target territory, summarized according to the standard climate classification of Europe. Included are the trend for the period 1950–2018 and the annual accumulated temperature, averaged for the periods 1950–1989 and 1990–2018, since a clear inflection point in the trend of temperature was noticed around the year 1990. The “% of change” indicates the difference of the accumulated annual temperature between the time series 1950–1989 and 1990–2018. AAT means for Annual accumulated temperature in °C.

Ecological Region	Trend 1950–2016	AAT: 1950–1989	AAT: 1990–2018	% Change Between 1950–1989 and 1990–2018
Alpine North	8.38	0.96	267.00	86.90
Alpine South	8.91	−2.19	65.32	26.29
Atlantic Central	7.63	3492.24	3745.04	7.25
Atlantic North	7.17	2871.40	3107.71	8.37
Boreal	10.66	742.51	1089.42	157.09
Continental	9.15	2890.30	3190.26	10.49
Nemoral	10.56	2135.75	2480.61	16.50

**Table 2 pathogens-09-00345-t002:** Trends in the four modelled developmental stages of the life cycle of *Ixodes ricinus* grouped according to the ecoregion in Europe (LANMAP2). OV: Oviposition. INC: Incubation. LN: larva to nymph. NA: nymph to adult. The negative trend means for a shortening of processes, the larger the negative number, the shorter the period.

Ecological Region	OV: Trend	INC: Trend	LN: Trend	NA: Trend
Alpine North	−0.69	−0.83	−0.89	−0.50
Alpine South	−0.42	−0.69	−0.78	−0.36
Atlantic Central	−0.31	−0.42	−0.47	−0.23
Atlantic North	−0.33	−0.46	−0.52	−0.25
Boreal	−1.35	−1.78	−1.94	−1.15
Continental	−0.42	−0.63	−0.72	−0.34
Nemoral	−0.69	−1.09	−1.25	−0.56

**Table 3 pathogens-09-00345-t003:** Changes in the four modelled developmental stages of the life cycle of *I. ricinus* grouped according to the ecoregion in Europe. OV: Oviposition. INC: Incubation. LN: larva to nymph molt. NA: nymph to adult molt. The table includes the predicted duration (days) of each stage according to the annual accumulated temperature in the periods 1951–1989 and 1990–2018. The second line in the columns of the period 1990–2018 means for the percent of difference between the time periods.

Ecoregion	OV: 1950–1989	INC: 1950–1989	LN: 1950–1989	NA: 1950–1989	OV: 1990–2018	INC: 1990–2018	LN: 1990–2018	NA: 1990–2018
Alpine North	96.14	90.06	90.59	86.48	70.6 (23.49%)	60.27 (31.47%)	58.95 (33.43%)	67.41 (21.05%)
Alpine South	111.36	120.12	123.39	109.44	94 (15.58%)	93.88 (21.84%)	94.07 (23.76%)	94.19 (13.93%)
Atlantic Central	101.75	91.06	91.59	89.38	88.17 (23.49%)	74.32 (18.03%)	73.22 (19.65%)	78.87 (11.59%)
Atlantic North	104.21	91.74	92.79	88.42	89.82 (13.79%)	73.8 (19.61%)	73.01 (21.35%)	77.34 (12.58%)
Boreal	137.41	146.55	146.20	129.28	89.63 (34.01%)	84.58 (42.55%)	79.26 (46.07%)	88.02 (31.06%)
Continental	107.76	105.78	107.94	98.78	90.43 (16.06%)	81.79 (22.76%)	81.29 (24.76%)	84.53 (14.48%)
Nemoral	116.08	124.86	130.47	108.02	89.89 (22.91%)	85.52 (32.18%)	85.99 (34.89%)	86.31 (20.33%)

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
