# Peer review of "A Retrospective Assessment of Temperature Trends in Northern Europe Reveals a Deep Impact on the Life Cycle of Ixodes ricinus (Acari: Ixodidae)"

_pathogens, 2020, doi:10.3390/pathogens9050345_

Round 1

Reviewer 1 Report

This manuscript by Estrada-Pena  has collated temperature data from the last 69 years (1950-2081) over regions of Europe and the impact of rising temperatures on the development rate of ixode ricinus ticks. This is an important area of study and of interest to a broad range of researchers.   the limitations of the study , as the authors have also acknowledged, is that it relies on temperature and does not take into account the changes in water vapor or humidity in the  atmosphere that would also impact the development stages. Perhaps, more so the mortality.   Despite the limitations, it is a well-thought out study .  However, presentation of the data and methodologies are not clear and one has to read it multiple time to understand. My specific comments are below:

The incubation period ? Does not clarify what is the incubation period.

The lack of data on humidity and host availability would make it difficult to evaluate the entire life cycle : not clear what they mean by that - do they mean mortality and density ?

How do changes in temperature affect host availability -and could this have a bigger impact on tick development ? This is not factored in the temperature -focused conclusion. 

Fig 3 . Are all aspects of development equally impacted ? It appears so from this figure - it is also not Discussed.

AAT needs to clarified and explained and this would make Figure 2 interpretable.

Minor points regards language : several places prepositions are missing  (for example : because its role instead of because of its role (line 34 ) -such errors are in multiple places 

Change  of temp ? it should be change in temperature (also in multiple places).

Author Response

This manuscript by Estrada-Pena  has collated temperature data from the last 69 years (1950-2081) over regions of Europe and the impact of rising temperatures on the development rate of ixode ricinus ticks. This is an important area of study and of interest to a broad range of researchers.   the limitations of the study , as the authors have also acknowledged, is that it relies on temperature and does not take into account the changes in water vapor or humidity in the  atmosphere that would also impact the development stages. Perhaps, more so the mortality.   Despite the limitations, it is a well-thought out study .  However, presentation of the data and methodologies are not clear and one has to read it multiple time to understand. My specific comments are below:

Dear Reviewer,

Many thanks for the constructive and candid criticisms. What follows is a point by point response to your questions:

The incubation period ? Does not clarify what is the incubation period.

- We included a clarification that it refers to the incubation of the eggs, since they are layed by the female until larvae hatch. In this in lines 276-277 (please note I refer to the lines o the Word dcouemtn. I do not know if the PDF produced by the web system could have a different line numbering).

The lack of data on humidity and host availability would make it difficult to evaluate the entire life cycle : not clear what they mean by that - do they mean mortality and density ?

- The Reviewer is right in pointing this detail because did a wrong choice of wording. The sentence now states “density and mortality”

How do changes in temperature affect host availability -and could this have a bigger impact on tick development ? This is not factored in the temperature -focused conclusion. 

- This is a very interesting question that, unluckily, has a not a straightforward response. We would need to model the many scopes of vertebrates that can act as hosts for the tick, resulting probably in a simple mapping exercise without “real” results. What I meant is that each vertebrate may have local and different impacts, additional modelling resulting in a pure “probabilistic exercise” and nothing else. We however think that the question is of great interest, and therefore we included new sentences about the topic in the Discussion.

Fig 3 . Are all aspects of development equally impacted ? It appears so from this figure - it is also not Discussed.

- No, they are affected in different ways. We think that the figures included in the Word document (as compulsory by the rules of the journal) is of too low quality as to capture all its meaning. However, a high resolution independent figure is available for displaying at high zoom in the computer’s screen. It shows that small and most probably not significant changes can be observed. We preferred no to address the topic, since evidence is weak.

AAT needs to clarified and explained and this would make Figure 2 interpretable.

- With our apologies if we misunderstood this comment, AAT is ONLY used in Table 1 and it is completely defined with the sentence “AAT means for Annual accumulated temperature in ºC.” (sic in the legend of Table 1). Figure 2 has a complete description including the units of the variables displayed. In any case, we believe this derives from the structure eo the papers in the Journal, with the Methods “after” the Results and the Discussion. We think we managed to fix these possible mistakes in the interpretation.

Minor points regards language : several places prepositions are missing  (for example : because its role instead of because of its role (line 34 ) -such errors are in multiple places 

Change  of temp ? it should be change in temperature (also in multiple places).

- We did all these necessary changes plus other, after an extensive reading of the complete manuscript. Many thanks for catching all these mistakes.

Reviewer 2 Report

The manuscript addresses an important issue regarding the relationship between climate change and tick-borne diseases. It is well written and the conclusions justified. However, interest in the manuscript would be improved by placing the data more in the context of disease transmission and seasonality. For instance, how might a faster rate of life stage development change (increase) interactions with humans? How much of an increase in human exposure might one expect from vector expansion? How might increased temperature change vector-reservoir host interactions. Not expecting data on these variables or interactions, but believe that the audience for this journal would appreciate more context for the data in the Discussion regarding transmission and enzootic cycles involving ticks.

Author Response

The manuscript addresses an important issue regarding the relationship between climate change and tick-borne diseases. It is well written and the conclusions justified. However, interest in the manuscript would be improved by placing the data more in the context of disease transmission and seasonality. For instance, how might a faster rate of life stage development change (increase) interactions with humans? How much of an increase in human exposure might one expect from vector expansion? How might increased temperature change vector-reservoir host interactions. Not expecting data on these variables or interactions, but believe that the audience for this journal would appreciate more context for the data in the Discussion regarding transmission and enzootic cycles involving ticks.

Dear Reviewer,

Many thanks for the constructive and candid criticisms. What follows is a point by point response to your comments. 

- The Reviewer indicated a very interesting, and even polemic, commentary. The issues is that commonly there is hard to find a direct translation of the impact of one single variable (temperature in our case) into transmission rates of pathogens to humans. Studies suggest that the contact rates between ticks and reservoirs are much more complex. We however think that this interesting comment merits a place in the Discussion of our manuscript. We specifically included several sentences in two different paragraphs (easily tracked in the revised version of the manuscript) to state these concerns. We however would like to kindly note that our study “only” points to a clear impact of the temperature eon the development rates of the tick, and this could be a major reason of concern regains human health. We pointed, too, the need to develop an adequate framework to address these issues that are not longer a “question of the future”. They are happening now.